plant nutrition; metalloprotein; metal homeostasis; zinc sensor; zinc transporter.

**Corresponding author**:
Ute Krämer.
Email: Ute.Kraemer@ruhr-uni-bochum.de

**Associate Editor:**
Ingo Dreyer

# Changing paradigms for the micronutrient zinc, a known protein cofactor, as a signal relaying also cellular redox state

Ute Krämer [ORCID]

Molecular Genetics and Physiology of Plants, Ruhr University Bochum, Bochum, Germany

## Abstract

The micronutrient zinc (Zn) is often poorly available but toxic when present in excess, so a tightly controlled Zn homoeostasis network operates in all organisms. This review summarizes our present understanding of plant Zn homoeostasis. In *Arabidopsis*, about 1,900 Zn-binding metalloproteins require Zn as a cofactor. Abundant Zn metalloproteins reside in plastids, mitochondria and peroxisomes, emphasizing the need to address how Zn reaches these proteins. Apo–Zn metalloproteins do not acquire $Zn^{2+}$ from a cytosolic pool of free cations, but instead through associative ligand exchange from Zn-buffering molecules. The importance of cytosolic thiols in Zn buffering suggests that, besides elevated Zn influx, a more oxidized redox state is also predicted to cause elevated labile-bound Zn levels, consistent with the suppression of a Zn deficiency marker under oxidative stress. Therefore, we consider a broadened physiological scope in plants for a possible signalling role of $Zn^{2+}$, experimentally supported only in animals to date.

## 1. Nutritional Zn requirements in plants and a narrow range of permissive Zn concentrations

Minimum required concentrations of the micronutrient zinc (Zn) in living tissues of land plants are around 30 μM, corresponding to 20 mg $kg^{-1}$ dry biomass (Krämer, 2010, 2024; Marschner & Marschner, 2012; Sinclair & Krämer, 2012; Stanton et al., 2022). Plants experience Zn toxicity generally above ~300 mg $kg^{-1}$ dry biomass (450 μM). Thus, the range of tissue Zn levels permissive for plant growth is narrow by comparison to most other mineral nutrients, although plants store considerable amounts of Zn in harmless form inside the vacuoles (usually ~100 μM, see below) and extracellularly bound to cell walls, where they are separated from the most sensitive metabolic pathways. Plants can face Zn deficiency because of its generally low abundance in soils, with totals between 10 and 300, and an average of around 55 mg Zn $kg^{-1}$ (Noulas et al., 2018). More importantly, $Zn^{2+}$ cations are poorly soluble, resulting in low Zn concentrations of ~1 μM in the soil solution from which plants take up nutrients (Krämer, 2024; Marschner & Marschner, 2012). When present in excess, Zn can cause toxicity directly, and also indirectly through nutrient imbalances arising from the competition between $Zn^{2+}$ and other divalent cations for uptake by roots and within the plant (Krämer, 2024). There are well-known examples of soils containing excess, toxic levels of Zn, and of Zn-deficient soils, as well as of plant species, ecotypes and crop varieties capable of growing on such soils (Alloway, 2009; Ernst, 1974; Marschner & Marschner, 2012). Moreover, even in common plants including the major reference species, researchers have characterized considerable Zn-related plasticity comprising responses to both Zn deficiency and Zn excess (Assunção, 2022; Sinclair & Krämer, 2012). This implies that all land plants encounter non-optimal Zn levels internally during their life cycle and externally in their roots' microenvironment. The latter is likely to reflect heterogeneity and dynamics in soil composition at multiple scales (Stark, 1994). In natural settings, the activities of symbiotic and associated microbes contribute to root Zn acquisition and the exclusion of excess Zn (Gonzalez-Guerrero et al., 2016). Most of our present knowledge of the involved molecular mechanisms was acquired by studying plants in isolation, and yet, it evidently has major predictive relevance also for plants cultivated in the soil in the presence of microbes (Sinclair & Krämer, 2012; Stanton

et al., 2022; Thiébaut & Hanikenne, 2022). In light of fluctuations in both environmental Zn availability and internal Zn demands, plant Zn homoeostasis has the roles of preventing Zn toxicity and of maintaining the functionality of all Zn-requiring proteins. A second, broader biological role of Zn homoeostasis in general $Zn^{2+}$-dependent signalling and regulation is discussed in this article, and a third role in mineral co-option (such as elemental defence) has recently been addressed elsewhere (Krämer, 2024).

## 2. Distinctive biological chemistry of the micronutrient Zn provided opportunities for evolutionary innovations

The transition metal element Zn, which is positioned in the d-block of the periodic table, occurs exclusively in the +II oxidation state in biology (Fraústo da Silva & Williams, 2001; Krezel & Maret, 2016). Adding on to the low solubility of many $Zn^{II}$ salts, such as phosphates and oxides, the $Zn^{2+}$ cation acts as a strong Lewis acid that establishes exceptionally stable high-affinity coordinative bonds in solution to form complexes with a large variety of ligands acting as free electron pair donors, by comparison to $Mg^{2+}$ and $Ca^{2+}$ (Box 1.1) (Fraústo da Silva & Williams, 2001). The so-called chelation of $Zn^{2+}$, which is coordinative $Zn^{2+}$ binding to the free electron pairs of several donor atoms of the same ligand molecule, can form particularly stable complexes – a characteristic of the divalent cations of transition metal elements, as well as of the alkaline earth metals $Mg^{2+}$ and $Ca^{2+}$. When a metal cation undergoes complex formation in the presence of a ligand, this decreases the 'chemically active' concentration of the metal cation interacting merely with surrounding water molecules ('free aqueous', 'hydrated'; $Me^{n+}_{aq}$, e.g., $Zn^{2+}_{aq}$). As a result, another additional ligand or binding partner will tend to bind lower net total concentrations of the metal at chemical equilibrium.

After binding to a ligand metabolite or a protein, the dissociation of $Zn^{2+}$ to form the hydrated cation occurs far more slowly than for $Ca^{2+}$, for example (dissociation rate constant $k_{off}$ is around five orders of magnitude smaller, based on approximated generalized rate constants) (Fraústo da Silva & Williams, 2001). $Zn^{2+}$ can displace not only $Ca^{2+}$ and $Mg^{2+}$ but even other transition metal micronutrient cations, such as $Fe^{2+}$, $Mn^{2+}$ and $Co^{2+}$, from solvent-exposed binding sites of proteins and bound ligands of any chemical nature, following the Irving–Williams series (Box 1.2) (Fraústo da Silva & Williams, 2001). The stability of Zn complexes is generally the largest for the binding of $Zn^{2+}$ involving thiol groups, followed by free electron pairs of nitrogen atoms and, subsequently, oxygen atoms contained in ligands (Williams, 2012). Moreover, compared to $Fe^{2+}$, $Mn^{2+}$ and $Co^{2+}$, the increase in stability of a complex formed with $Zn^{2+}$ is generally the largest for thiol, intermediate for nitrogen, and the smallest for oxygen coordination (Williams, 2012).

As a strong Lewis acid, $Zn^{2+}$ is highly effective in polarizing bound ligands (Box 1.1). Its preferred coordination geometry is tetrahedral, but more flexible than that of other transition metal cations (Krezel & Maret, 2016; Stanton et al., 2022). Whereas dissociative ligand exchange via a free hydrated aqueous $Zn^{2+}$ cation ($Zn^{2+}_{aq}$) intermediate occurs exceedingly slowly, associative ligand exchange, that is an exchange of ligands directly on the metal without any formation of $Zn^{2+}_{aq}$ as an intermediate, can occur rapidly, in the order of $10^6$ times faster (Box 1.3) (Costello et al., 2011; Foster et al., 2014; Heinz et al., 2005; Krezel & Maret, 2016). Thus, associative ligand exchange is now thought to be of central

relevance also for the integration of Zn into apo-metalloproteins (protein metalation) and not only for Zn-mediated catalysis *in vivo*. In summary, the biological chemistry of $Zn^{2+}$ is distinct from that of other nutrient and non-nutrient transition metal cations, and it differs decisively from that of alkali and alkaline earth metal cations including $Ca^{2+}$.

---

**Box 1. Glossary**

1. **Lewis acid**: electrophilic ('electron-loving') acceptor ion or molecule capable of attaching itself to an electron pair of a donor ion or molecule (Lewis base)

2. **Irving–Williams series** of increasing stability of complexes with ligands[i]:
$$Ca^{2+}/Mg^{2+} < Mn^{2+} < Fe^{2+} < Co^{2+} < Ni^{2+} < Cu^{2+} > Zn^{2+}$$

3. **Dissociative** ligand exchange versus **associative** ligand exchange on $Zn^{2+}$:
   (a) DISSOCIATIVE: $P + [ZnX] \leftrightarrow P + Zn^{2+}_{aq} + X \leftrightarrow [PZn] + X$
   (b) ASSOCIATIVE: $P + [ZnX] \leftrightarrow [PZnX] \leftrightarrow [PZn] + X$

   (X: ligand; P: target protein; chemists group the components of a metal complex using square brackets)

4. **Zn metalloprotein types**[ii] (classification suggested by Krämer, 2024):

   **Type I Zn metalloprotein**: depends on bound $Zn^{2+}$ acting as a structural or catalytic cofactor for its biological function as an enzyme or transcription factor, for example (but excluding Type II)

   **Type II Zn metalloprotein**: acting directly in metal homoeostasis and functioning to reversibly bind $Zn^{2+}$ or a Zn-containing complex, with roles in the transmembrane transport, storage, detoxification, mobilization, immobilization, movement, shuttling, transfer or the acquisition of Zn, for example

   **Type III Zn metalloprotein**: regulatory switch capable of $Zn^{2+}$ binding and dissociation, thus toggling between two opposing functional states (a $Zn^{2+}$-bound and a $Zn^{2+}$-free state), which activates or deactivates a biological process or a signalling cascade

5. **Metal buffer** and **buffered available $Zn^{2+}$ concentration**: $Zn^{2+}$ and all other cations that have similar binding properties (predominantly transition metal cations), as well as all ligands and solvent-accessible metal-binding sites capable of reversibly binding $Zn^{2+}$ or other metals provided by (a) low-molecular-mass organic and inorganic compounds, and the surfaces of (b) proteins and other macromolecules or structures present in the cytosol or the chloroplast stroma, for example
   (a) glutathione (GSH); free amino acids (glutamate, glutamine, histidine and others); organic acids (citrate, malate); nicotianamine; phytochelatins; ATP; inositol phosphates (including phytate); other phosphates and diphosphates[iii], for example
   (b) metallothioneins as specialized $Zn^{2+}$-buffering proteins, all surface-exposed metal-binding sites of proteins, of membrane surfaces, and of carbohydrates, for example

   The **buffered available $Zn^{2+}$ concentration** $[Zn^{II}_b]$ corresponds only formally to a concentration of **free aqueous $Zn^{2+}$ ($Zn^{2+}_{aq}$)**. It reflects the free energy that quantifies the availability of $Zn^{2+}$ to bind to another ligand.

   **i** Binding/stability constant $K_s = k_{on}/k_{off}$; the dissociation constant $K_d = K_s^{-1}$; $pK_d = -Log_{10}K_d$.
   **ii** Different from here, $Zn^{2+}$ as used here is often referred to as 'Zn', 'Zn(II)' or '$Zn^{II}$'; and '$Zn^{2+}$' is often used to refer to $Zn^{2+}_{aq}$ elsewhere in plant biology and biochemistry texts.
   **iii** Phosphates and free aqueous metal cations are prone to forming precipitates, which are no longer quantitatively part of the metal buffer. Organisms are likely to keep precipitation at a minimum at cellular sites of protein metalation outside storage sites in seeds and in (part of the) vacuoles, for instance.

## 3. The functions of Zn in the plant metalloproteome

Of the 27,500 protein-coding loci of *Arabidopsis*, about 1,900 proteins (7%) were estimated to be Zn metalloproteins (Andreini et al., 2006, 2009; Zhang & Krämer, 2018) – a list that has to be considered preliminary. These proteins comprise enzymes that depend on $Zn^{2+}$ as a catalytic cofactor participating directly in substrate conversion or as a structural cofactor conferring the functional three-dimensional tertiary structure of a protein (Type I metalloproteins; Box 1.4). Proteins mediating transmembrane Zn transport or Zn binding directly in the operational maintenance of plant Zn homoeostasis are also addressed as Zn metalloproteins (Type II; Box 1.4). In comparison to metals with similar chemical properties, $Zn^{2+}$-mediated catalysis is faster and is more versatile, thus explaining the use of Zn as a cofactor of enzymes across all functional classes (Fraústo da Silva & Williams, 2001). Its high binding affinities and the absence of redox transitions under biologically relevant conditions may explain the exceptionally widespread use of $Zn^{2+}$ as a structural cofactor, such as in Zn finger domains which often mediate protein–DNA or protein–protein interactions.

Comparisons among organisms suggest an increasing use of $Zn^{2+}$ with evolution, from 5% to 6% of bacterial and archaeal proteomes to around 9% in eukaryotes (Andreini et al., 2009). Early during the evolution of life, Zn was immobilized in the form of insoluble sulphides and, thus, was unavailable. With $\sim10^8$-fold increased levels of Zn (to around 10 nM) in the oceans upon the oxygenation of the Earth's atmosphere, organisms apparently evolved to gradually replace other metals, in particular $Ni^{2+}$, $Co^{2+}$ and $Fe^{2+}$, by $Zn^{2+}$ as a catalytic cofactor in a variety of enzymatic functions (Fraústo da Silva & Williams, 2001; Krämer, 2024). In addition, novel protein functions in transcriptional regulation and protein–protein interactions increasingly made use of Zn in the evolution of eukaryotes and multicellular organisms, alongside an overall elevated bioavailability of Zn on land. Compared to our knowledge of type I and type II metalloproteins, which remains sketchy and relies heavily on *in silico* predictions, our knowledge of type III Zn metalloproteins ($Zn^{2+}$ sensors; Box 1.4) in plants is even less developed because of their poor conservation across the kingdoms of life (Andreini et al., 2006, 2009; Krämer, 2024). Type III metalloproteins are characterized by the presence of facultative metal binding sites, which confer alternate protein functional states in the metal-bound and -unbound forms. Important examples of $Zn^{2+}$ sensors in plants are the basic leucine zipper proteins AtbZIP19 and AtbZIP23, two partly redundant activators of the transcription of genes encoding $Zn^{2+}$ transporters and chelator biosynthetic enzymes constituting a subset of the known transcriptional Zn deficiency responses (Assunção et al., 2010; Lilay et al., 2021) (Figure 1a). These proteins contain a $Zn^{2+}$-binding motif in the N-terminal region preceding the DNA-binding motif and the leucine zipper, and only their apo-form is capable of activating transcription (Assunção, 2022; Lilay et al., 2018). In addition, the root high-affinity Fe uptake protein iron-regulated transporter 1 (IRT1) could be considered another example of a type III Zn metalloprotein. The binding of $Zn^{2+}$, which is a secondary substrate of the transporter, to a cytosolic loop of IRT1 triggers its own phosphorylation, polyubiquitination-mediated removal from the plasma membrane and degradation (Dubeaux et al., 2018).

Of the 1,921 Zn metalloproteins (Zhang & Krämer, 2018) identified based on known Zn-binding motifs (Andreini et al., 2009), proteomics of *Arabidopsis* rosette leaves identified 636 (Mergner et al., 2020a), as can be established by comparing published datasets. Furthermore, approximated relative numbers

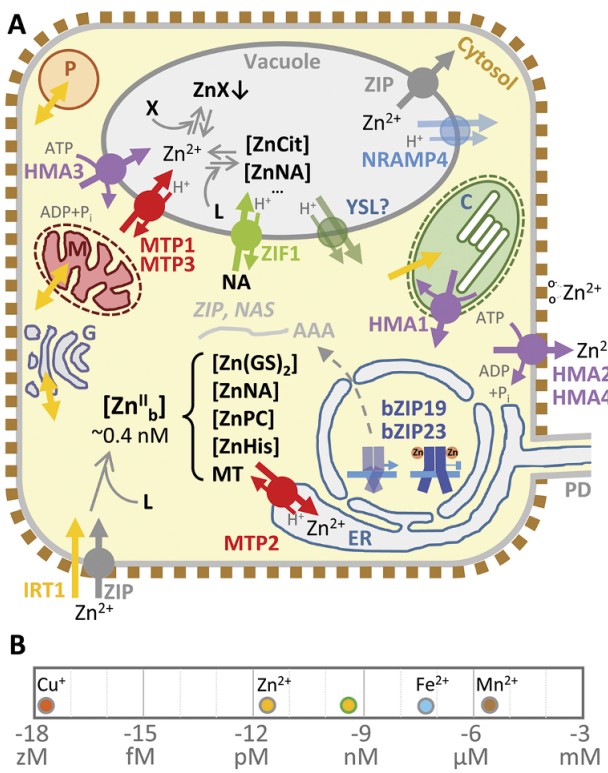

**Figure 1. Cellular zinc homoeostasis in plants.** (a) Cellular Zn homoeostasis and the cytosolic Zn buffer. (b) Cytosolic buffered available metal cation concentrations in *Salmonella enterica* (grey line around symbols) and in *Arabidopsis thaliana* (for $Zn^{2+}$ only, green line around symbol). $[Zn^{II}_b]$, the buffered available $Zn^{2+}$ concentration, is dependent on the metal buffer. Cytosolic Zn is present in the form of Zn complexes of low-molecular-mass ligands, such as glutathione (GSH) to form $[Zn(GS)_2]$, nicotianamine ($[ZnNA]$), phytochelatins ($[ZnPC]$), histidine ($[ZnHis]$) (ligand biosynthesis not shown), as well as Zn-binding sites of type I Zn metalloproteins (not shown) and metallothionein (MT) (a). Dashed arrow indicates the movement of mRNAs (drawn line ending in AAA, with names) of bZIP19/23 target genes out of the nucleus under Zn deficiency (a). C, plastid; Cit, citrate; ER, endoplasmatic reticulum; G, Golgi apparatus; L, ligand forming complex; M, mitochondrium; P, peroxisome; PD: plasmodesma; X, phosphate or other anion forming Zn-precipitates. Yellow stripe-like (YSL) and natural resistance associated macrophage protein (NRAMP) family proteins can transport Zn, but critical roles in Zn homoeostasis have not yet been demonstrated for them in *Arabidopsis*. They might alternatively mediate Zn influx into the cytosol across other membranes, such as the plasma membrane. See text for other abbreviations and descriptions; more details have been described elsewhere (Sinclair & Krämer, 2012).

of Zn-metalloprotein molecules were about 10%–20% higher in the oldest leaf than in the youngest leaf. According to proteomics-based abundance estimates, carbonic anhydrases, proteases, Calvin–Benson–Bassham cycle and central metabolism enzymes were among the top 15 most abundant Zn metalloproteins in leaves, with an apparent predominance of plastid-localized proteins (Table 1, similar when using leaf data from Mergner et al., 2020b). Cytosolic ribosomal Zn metalloproteins were among the most abundant proteins in pollen, roots and the shoot apical meristem, for example (Mergner et al., 2020b).

## 4. Protein metalation: how $Zn^{2+}$ cations reach their cognate binding sites

Given the comparably high binding affinity of $Zn^{2+}$ for a variety of ligands, the questions arise of how it is incorporated (exclusively) in

**Table 1.** The top 15 most abundant Zn proteins in leaves of *Arabidopsis*

| AGI code | Localization[a] – Description | Short | Leaf 1 (oldest) | Leaf 4 | Leaf 12 |
|---|---|---|---|---|---|
| | | | $[10^6 \times \text{TMT}]$[b] | | |
| AT5G14740 | pl – CARBONIC ANHYDRASE 2 | CA2 | 4.0 | 4.3 | 2.0 |
| AT5G17920 | cyt – COBALAMIN-INDEPENDENT METHIONINE SYNTHASE | ATMS1 | 1.5 | 1.7 | 2.1 |
| AT3G55800 | pl – SEDOHEPTULOSE–1,7-BISPHOSPHATASE[c,d] | SBPASE | 2.4 | 2.2 | 1.2 |
| AT2G30950 | pl – VARIEGATED 2, FTSH metalloprotease 2 | VAR2 | 1.5 | 1.5 | 1.2 |
| AT3G01500 | pl – CARBONIC ANHYDRASE 1 | CA1 | 1.3 | 1.4 | 0.68 |
| AT1G50250 | pl – FTSH metalloprotease 1 | FTSH1 | 1.3 | 1.3 | 0.90 |
| AT1G70730 | cyt – PHOSPHOGLUCOMUTASE 2[d] | PGM2 | 1.5 | 1.4 | 0.76 |
| AT1G63770 | pl – Peptidase M1 family protein | | 1.0 | 1.0 | 0.87 |
| AT3G54050 | pl – HIGH CYCLIC ELECTRON FLOW 1, fructose 1,6-bishosphate phosphatase[c,d] | HCEF1 | 1.2 | 1.2 | 0.63 |
| AT5G65620 | pl&mt – ORGANELLAR OLIGOPEPTIDASE[e] | OOP | 0.9 | 1.0 | 0.76 |
| AT5G43940 | per – *S*-NITROSOGLUTATHIONE REDUCTASE | GSNOR1 | 1.0 | 1.0 | 0.68 |
| AT5G26742 | pl – DEAD box RNA helicase (emb1138)[f] | RH3 | 0.48 | 0.50 | 1.4 |
| AT4G33090 | gol&pm - AMINOPEPTIDASE M1 | APM1 | 0.91 | 0.92 | 0.74 |
| AT5G51820 | pl – PHOSPHOGLUCOMUTASE, STARCH-FREE 1 (STF1) | PGM1 | 0.76 | 0.79 | 0.54 |
| AT2G24200 | cyt – LEUCYL AMINOPEPTIDASE 1 | LAP1 | 1.2 | 1.1 | 0.57 |

[a] https://suba.live/; cyt, cytosol; gol, Golgi apparatus; mt, mitochondria; per, peroxisomes; pl, plastids; pm, plasma membrane.
[b] Numbers are mio. TMT (tandem mass tags), as a preliminary approximation of abundance (Mergner et al., 2020b).
[c] Calvin–Benson–Bassham Cycle.
[d] Central metabolism.
[e] Kmiec et al., (2013).
[f] Splicing of Group II introns, relevant for chloroplast ribosome.

its cognate binding sites of newly synthesized apo-metalloproteins and how the cell prevents the binding of other metals, in particular Cu (Box 1.2). Extrapolating from studies in bacteria and mammalian cells, our present concept of protein metalation in plants is now based entirely on associative $Zn^{II}$ transfer (see above; Box 1.3) (Costello et al., 2011; Foster et al., 2022b; Krämer, 2024). A Zn-trafficking metallochaperone protein, in analogy with the role of Cu-metallochaperones in the metalation of apo-Cu-metalloproteins, is only known for one single Zn-dependent protein of plants to date (Pasquini et al., 2022). Instead, the metalation of Zn-dependent apo-metalloproteins is now thought to occur co- or post-translationally from the cellular metal buffer via rapid ligand exchange reactions (Box 1.5) (Foster et al., 2022b; Krezel & Maret, 2016). The metal buffer consists of labile-bound $Zn^{II}$ and other transition metals and an overall excess of ligands and solvent-accessible metal-binding sites provided by a complex mix of metabolites, inorganic compounds, protein surfaces and other macromolecules, which can chelate $Zn^{2+}$ (Figure 1a) (Foster et al., 2022b; Osman & Robinson, 2023). In bacteria, intracellular buffered available $Zn^{2+}$ concentrations are thus maintained in the femtomolar to picomolar range (2 pM in aerobically cultivated *Escherichia coli*), which allows the adequate metalation of Zn and other metalloproteins (Box 1.5, Figures 1b and 2a) (Foster et al., 2022b; Outten & O'Halloran, 2001). This implies that there are no free hydrated $Zn^{2+}$ ions in the bacterial cytoplasm statistically, despite total intracellular Zn concentrations in the sub-millimolar range (Foster et al., 2022a; Outten & O'Halloran, 2001). In *Arabidopsis*, total intracellular Zn concentrations are thought to be at similar levels and cytosolic buffered available $Zn^{2+}$ levels were reported at 400 pM under standard conditions and 2 nM under excess-Zn conditions – at least 100-fold higher than in bacteria and similar to levels around 100 pM reported

in mammalian cells (Figure 1a, b) (Krämer, 2024; Lanquar et al., 2014; Qin et al., 2013). However, the availability of only a single published study on a land plant to date, and the use of different methods limit the comparability with bacterial studies. Plant vacuoles, which contain ~100 μM Zn, constitute a large potential cellular reservoir for the dynamic entry of metal ions into, and removal from, the cytosolic metal buffer – a feature that differs strongly from many other organisms (Lanquar et al., 2014, 2010). Experimental procedures involving the destruction of plant cells inevitably permit the redistribution of metal cations among all accessible binding sites in aqueous suspensions or solutions and are thus unable to provide any information on Zn localization, binding or speciation *in vivo*.

## 5. Metal buffering in plant cells highlights that a more oxidized redox state can generate an internal burst in available $Zn^{2+}$

With low-millimolar concentrations, a major contribution of glutathione (GSH), γ-glutamyl-cysteinyl-glycine (γ-ECG), to the cytosolic metal buffer appears feasible (Figures 1a and 2a) (apparent $pK_d$ of the $Zn(GS)_2$ complex ~8.1 at pH 7.7; Box 1.2) (Koffler et al., 2013; Krezel & Maret, 2016; Meyer et al., 2001). Plant metallothionein proteins, a family of small cysteine-rich proteins capable of metal binding, and other proteins containing surface thiols are likely to make notable contributions to the cytosolic metal buffer (Hassinen et al., 2011; Hübner & Haase, 2021). As a result of the preference of Zn for thiol-containing ligands including GSH, they are thought to contribute overproportionately to the intracellular buffering of $Zn^{2+}$, compared to the buffering of $Fe^{2+}$ and $Mn^{2+}$, for example. We would expect that any decrease in total

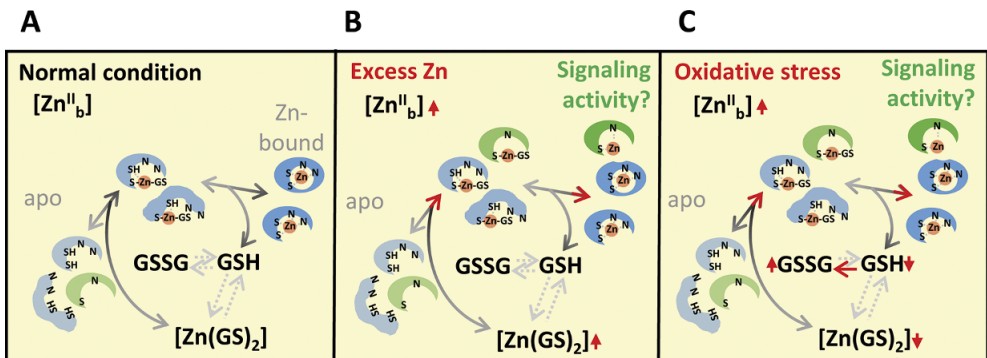

**Figure 2. Model of the metalation of Zn metalloproteins.** (a–c) Expectations of cytosolic protein metalation under normal conditions (a), excess Zn conditions (b), and conditions of a more oxidized redox state, for example, under oxidative stress (c). Apo-metalloproteins (left) undergo metalation from the Zn buffer by associative ligand exchange (middle, shown for the ligand GSH as an example) to generate the Zn-bound forms (right, blue shapes) (a–c). The darker end of the arrows and the progressively darker shapes indicate the direction during metalation, whereby each step is reversible; important equilibria are emphasized by dotted arrows (a–c). Protein conformational changes can render the bound Zn solvent-inaccessible, removing it from the buffer (upper right blue shape in a, shown again in b and c). Red vertical arrows show the direction of changes when cytosolic Zn levels increase (b), and when cytosolic GSH is oxidized to GSSG (c), and red arrowheads emphasize that metalation processes are generally favoured (b, c). This is predicted to result in the binding of $Zn^{2+}$ to labile binding sites on additional proteins (green shapes) including type III metalloproteins which can activate signalling and regulation (b, c). Note that a more oxidized redox state can also result in the formation of disulfide bridges among thiol groups of Zn-binding sites, thus strongly decreasing their affinity for $Zn^{2+}$ (not shown).

cytosolic thiol concentrations, either through their oxidation or through a decrease in total thiol levels, leads to increased buffered available cytosolic concentrations of $Zn^{2+}$, similar to a situation in which excess Zn levels are present in the cytosol (Box 1.5, Figure 2a–c). Under conditions of oxidative stress, we would thus expect at least a transient homoeostatic downregulation of known transcriptional Zn deficiency markers in *Arabidopsis*. Concordant with the transition to a more sufficient cellular Zn status, transcript levels of the bZIP19/bZIP23 target gene *ZIP3* are downregulated in roots of wild-type *Arabidopsis* plants after 2 h of oxidative stress (Figure 1a) (Lehmann et al., 2009), as well as in GSH-deficient mutants by comparison to the wild type (Schnaubelt et al., 2015). This highlights the potential relevance of changes in redox state for the homoeostasis of Zn in plants. Beyond this, transient increases in intracellular buffered available $Zn^{2+}$ concentrations could act as a signal contributing to the regulation of a broader range of cellular processes by altering the functional state of hitherto unidentified type III metalloproteins through the binding of $Zn^{2+}$ to facultative binding sites (Box 1.4, Figure 2a–c). This possibility expands the physiological scope for possible $Zn^{2+}$-dependent signalling, which could be triggered in plant cells through several processes. As is well-known, it could be initiated by the increased influx of Zn into the cytosol upon release from the vacuole or a sudden increase in extracellular bioavailable Zn levels (Figures 1a and 2b). Alternatively, it could be activated during episodes of a more oxidized redox state, which can result from cellular stress, dynamic changes in metabolism, or as a response to a variety of abiotic or biotic stimuli (Mittler et al., 2022) (Figure 2c). Beyond this, also a decrease in pH, predicted to result in the protonation of free electron pairs of important Zn ligands including GSH, could potentially lead to an increase in $Zn^{2+}$ availability. Despite a growing body of literature in the animal field on signalling and regulatory roles of $Zn^{2+}$ outside maintaining metal homoeostasis alone, they are yet to be experimentally examined in plants (Assunção, 2022; Clemens, 2022; Earley et al., 2021; Hübner & Haase, 2021; Krämer, 2024; Maret, 2019).

Based on the stability constants of Zn complexes of phytochelatins (PCs), which increase with increasing lengths of the ($\gamma$-EC)$_n$G peptide ($n$ = 2–5), it was proposed that PC2 (($\gamma$-EC)$_2$G)

contributes to the cytosolic metal buffer in plants cultivated under standard conditions (in the absence of a metal excess) (Luczkowski et al., 2024). The authors proposed that the stability of Zn complexes with PC2 (apparent p$K_d$ ~7.3 at pH 7.4; Box 1.2), with PC2 levels of 5–30 nmol g$^{-1}$ fresh biomass (i.e., up to around 30 μM in fresh tissues) in *Arabidopsis* roots, could contribute to cytosolic Zn buffering. Nicotianamine concentrations in *Arabidopsis* tissues are in a similar range as PC2 levels, and NA is thought to contribute to the cytosolic metal buffer (ca. 25–140 nmol g$^{-1}$ fresh biomass (Klatte et al., 2009; Pianelli et al., 2005; Weber et al., 2004), p$K_d$ = 15.4 (von Wirén et al., 1999) and apparent p$K_d$ ~10.48 at pH 7.7 (Krezel & Maret, 2016)). Yet, we are presently lacking important information on the cytosolic concentrations of these low-molecular-mass chelators and their subcellular distribution between the cytosol, the vacuole and intracellular vesicles, for instance (Chao et al., 2021; Haydon et al., 2012; Song et al., 2010).

Plant cells must also accomplish and maintain protein metalation in cellular compartments other than the cytosol. Work in mammalian model systems suggested buffered available $Zn^{2+}$ concentrations of 0.3 nM in mitochondria and >5 nM in the endoplasmatic reticulum (ER) (Chabosseau et al., 2014; Clemens, 2022). The latter is consistent with a more oxidized redox state of GSH in the ER than in the cytosol of plant cells (Schwarzländer et al., 2008). Under Zn deficiency, the possibility of maintaining higher buffered available $Zn^{2+}$ concentrations in the ER may help to maintain a rate of root-to-shoot Zn flux critical for survival (see below).

## 6. A brief overview of Zn acquisition and homoeostasis in *Arabidopsis*

The uptake of Zn into the root symplast of *Arabidopsis* and other plants is thought to occur via redundantly acting $Zn^{2+}$ transporters of the Zn-regulated transporter, IRT-like protein (ZIP) family, which are capable of mediating cellular Zn uptake when produced in the yeast *Saccharomyces cerevisiae* (Figure 1a) (Grotz et al., 1998; Lee et al., 2021). Yet, this model is not unequivocally supported by reverse genetic evidence. An increased vacuolar

accumulation of Zn in a quadruple *irt3 zip4 zip6 zip9* mutant and *irt3 zip4 zip6/zip9* triple mutants of *Arabidopsis thaliana* implies additional or other transporters in root Zn uptake, and it could alternatively reflect a role of these transporters in the remobilization of Zn from pools stored in the vacuoles of root cortex cells (Lee et al., 2021; Robe et al., 2024). Additional unidentified Zn transporters of plants might have dual substrates or transport Zn complexes. For example, the high-affinity inorganic phosphate uptake system PHO84 of *S. cerevisiae* was proposed to additionally mediate the uptake of Mn, Zn and other metals (Jensen et al., 2003). The closest plant homologues of *Sc*PHO84 are in the phosphate transporter family of major facilitator superfamily proteins (Nino-Gonzalez et al., 2019).

Zinc transport into vacuoles is mediated by metal transport/tolerance protein 1 (MTP1) and MTP3 of the so-called cation diffusion facilitator family, as well as heavy metal ATPase 3 (HMA3; note that Col-0 is an *hma3* mutant) (Figure 1a) (Arrivault et al., 2006; Desbrosses-Fonrouge et al., 2005; Morel et al., 2009). *MTP1* appears to be highly expressed in growing tips, whereas both *MTP3* and *HMA3* are upregulated in the roots of plants exposed to Fe deficiency and excess Zn. Transport of nicotianamine into the vacuoles via zinc-induced facilitator 1 (ZIF1) enhances vacuolar Zn storage and can contribute to immobilizing Zn in the roots (Haydon & Cobbett, 2007; Haydon et al., 2012). The function of ZIF1 illustrates the principle of 'Zn trapping' through the localized accumulation of a chelator, which can have a major impact on Zn partitioning within the plant. With similar effects, silicates, phosphates and polyphosphates such as phytate (inositol hexakisphosphate, IP6), for example, have been implicated in the formation of Zn precipitates, mostly extracellularly and inside vacuoles (Bouain et al., 2014; Neumann & zur Nieden, 2001).

*ZIF1* transcript levels respond in a similar manner as those of *MTP3* and *HMA3*. Root cortex cells appear to be important in Zn storage and immobilization based on the localization of promoter activities of *HMA3*, *MTP3* and *ZIF1*. Only the Fe deficiency responsiveness of *HMA3* and *MTP3*, but not that of *ZIF1* transcript levels, depends on upstream regulator of IRT1 (URI) and Fe deficiency-induced transcription factor 1 (FIT1), central transcriptional activators of Fe deficiency responses (Colangelo & Guerinot, 2004; Kim et al., 2019; Mai et al., 2016). The similarities between the responses of roots to excess Zn and Fe deficiency highlight the ability of Zn to interfere with Fe nutrition and the relevance of the high-affinity Fe uptake system IRT1 for the influx of $Zn^{2+}$ into root cells as a secondary substrate (Figure 1a). Consistent with the disruption of Fe nutrition by exposure to excess Zn, *brutus-like 1* (*btsl1*) *btsl2* double mutants are more Zn-tolerant than wild-type *Arabidopsis* plants. This has been attributed to aspects of a general upregulation of Fe deficiency responses in these mutants (Stanton et al., 2023).

Nicotianamine, when not sequestered in vacuoles, appears to be important in the radial symplastic cell-to-cell movement of Zn in roots towards the stele, where $Zn^{2+}$ cations are exported into the xylem by the metal pumps HMA4 and HMA2 (Figure 1a) (Deinlein et al., 2012; Haydon et al., 2012; Hussain et al., 2004). These two metal pumps act partially redundantly. Whereas *HMA4* transcripts are constitutively present in *A. thaliana* roots at low levels, the levels of *HMA2* transcripts are upregulated under Zn deficiency (Sinclair et al., 2018). Compared to *A. thaliana*, strongly elevated *HMA4* transcript levels in *Arabidopsis halleri* are necessary for naturally selected Zn hyperaccumulation, the accumulation of extraordinarily high Zn levels in shoots of this species (Hanikenne et al., 2008; Krämer, 2010). The high rates of HMA4-dependent

$Zn^{2+}$ export from the root symplasm into the xylem were shown to cause the secondary homoeostatic upregulation of the levels of Zn deficiency-responsive bZIP19/bZIP23 target transcripts in roots of *A. halleri*, presumably as a result of Zn depletion in root cells (Hanikenne et al., 2008). This is a second example of how changing a single process in the Zn homoeostasis network can strongly affect Zn fluxes in plants (see ZIF1 above).

ZIP family transporters are thought to contribute to xylem unloading in the shoot by mediating Zn uptake into the adjacent cells. The molecular mechanisms involved in the distribution of Zn across the leaf blade are poorly understood. HMA4 is likely to contribute to this at least in *A. halleri*, as well as in *A. thaliana* under specific conditions such as pathogen attack, in the context of elemental defences against biotic stress (Escudero et al., 2022; Hanikenne et al., 2008; Krämer, 2024).

Zn deficiency responses are regulated in a cell-autonomous manner by the Zn-sensing transcription factors bZIP19 and bZIP23 (Figure 1a), which may also act as heterodimers and have partially redundant functions (Assunção, 2022; Kimura et al., 2023; Sinclair et al., 2018). Targets with known roles in Zn homoeostasis are primarily subsets of ZIP transporter- and nicotianamine synthase-encoding genes (Assunção et al., 2010). In addition, there is a systemically regulated Zn deficiency response in roots controlled by the Zn status of rosette leaves, which includes the upregulation of transcript levels of *MTP2* and *HMA2* (Sinclair et al., 2018). Both of the encoded proteins contribute to enhancing root-to-shoot Zn flux. It appears that MTP2 acts by transporting $Zn^{2+}$ into the ER (Figure 1a), suggesting that radial cell-to-cell transport of Zn in roots occurs within desmotubules under Zn deficiency, which will require further study.

Both HMA2 and HMA4 also act in the export of Zn from the maternal seed coat for its translocation to the developing embryo (Olsen et al., 2016). *MTP2* promoter activity and transcripts are also found in developing seeds of Zn-deficient plants. In addition to ZIP family proteins, natural resistance associated macrophage protein (NRAMP) and yellow stripe-like (YSL) family proteins are candidate membrane transporters for contributing to the remobilization of $Zn^{2+}$ from vacuoles or other cellular compartments (Figure 1a), and also for mediating Zn influx across the plasma membrane. HMA1 is thought to export Zn from the chloroplast stroma to protect chloroplasts from Zn toxicity (Figure 1a) (Kim et al., 2009). Although a number of critical Zn metalloproteins localize to plastids and mitochondria, we do not presently know through which transporters Zn enters these organelles.

Basal Zn tolerance (Clemens, 2001) involves Zn-binding PCs (($\gamma$-EC)$_n$G, $n$ = 2–11). PCs are synthesized from GSH by PC synthase enzymes, which are activated by metal binding (Tennstedt et al., 2009). The proposed roles of cysteine-rich extracellular plant defensins in Zn-binding and Zn tolerance in plants are less well understood (Shahzad et al., 2013). Apoplastically localized cysteine-rich defensin-like (DEFL) proteins have been implicated in root Zn deficiency responses (Kimura et al., 2023; Niehs et al., 2024). *DEFL202*, *203* and *206–208* are transcriptionally upregulated under Zn deficiency and – different from the wild type – *defl202 defl203* double mutants fail to attenuate root growth under Zn deficiency (Kimura et al., 2023).

*Arabidopsis de-etiolated by zinc* (*dez*, or *trichome birefringence*, *tbr*) and *overly zinc sensitive 2* (*ozs2*, or *pectin methylesterase 3*, *pme3*) mutants are Zn hypersensitive, and this was demonstrated to be related to reduced contents or accessibility of de-methylesterified pectin in the cell wall (Sinclair et al., 2017; Weber

et al., 2013). The extracellular binding of $Zn^{2+}$ to cell walls thus contributes to basal Zn tolerance. Recently identified natural variants of *TBR* may relate to this, but we presently do not know whether this natural genetic variation is associated with variation in bioavailable Zn levels in the soils of origin of these *Arabidopsis* accessions (Zhong et al., 2024). The current status of knowledge suggests that the main proteins and processes of Zn homoeostasis are conserved across flowering plants, including also rice, for example (Stanton et al., 2022).

In summary, there is plenty of scope for research towards an improved understanding of Zn homoeostasis in plants. Future research will address the nature, abundance and subcellular localization of Zn metalloproteins, protein metalation, the cellular dynamics of Zn availability, as well as aim to identify $Zn^{2+}$-sensing proteins and their roles in $Zn^{2+}$-dependent signalling and regulation, both within and beyond Zn homoeostasis alone. In addition to the role of Zn in its metalloproteome, Zn can be co-opted for other functions such as elemental defence (see Krämer, 2024). Scientific progress in plant Zn biology will contribute to increasing yield, quality and stress resilience of crops.

**Data availability statement.** The contents of this publication are based on published data, and the corresponding references are cited. No coding was involved.

## Acknowledgements

The author would like to thank all colleagues in the metals field for numerous discussions and an intense exchange of ideas and opinions over many years. The author would also like to thank three anonymous reviewers for reading the manuscript carefully and for making thoughtful suggestions. The author would like to apologize to all colleagues whose work is not cited here due to space restrictions.

**Author contributions.** Ute Krämer conceived and designed the text, tables and figures, analysed data and wrote the article.

**Funding statement.** This work received funding from the European Research Commission (ERC-AdG 788380 LEAP EXTREME); the German Research Council (Deutsche Forschungsgemeinschaft, TRR 341 Plant Ecological Genetics grant No. 456082119, RTG 2341 MiCon grant No. 321933041 and Individual Research Grant No. 5453631); as well as financial support for exchange and discussion from the European Cooperation in Science and Technology (Cost Action PLANTMETALS, CA19116).

**Competing interest.** The author declares no competing interests.

**Open peer review.** To view the open peer review materials for this article, please visit http://doi.org/10.1017/qpb.2025.4.

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
