## [Reviewer Report]

The review by Ute Krämer provides an expert view on the mechanisms of zinc homeostasis in plants. The review contains many reference values and a introduces a quantitative analysis of the zinc proteome. It briefly recapitulates the knowledge on the molecular mechanisms of metal homeostasis. Starting from basic physicochemical principles and well-established data in prokaryotic cells, the author proposes inspiring new hypotheses about the involvement of glutathione in zinc buffering and a role for zinc as a signal in plant cells. The review thus establishes a framework for future research in this field.

The manuscript is written in a clear and concise style. I have only very few suggestions for optimization:

1) The sections “protein metalation, binding and unbinding of cellular zinc ions” and “how is zinc buffered in plant cells” are somewhat redundant. Both sections highlight the probable role of glutathione in zinc buffering. As the two processes are intimately linked, would it be possible to merge the two sections to avoid redundancies?

2) The review does not address the interactions between zinc and phosphate or polyphosphate, which might be relevant for buffering and transport.

3) It would be nice to provide values for:

- “considerable amounts of zinc in harmless form inside the vacuole” (concentration, line 53)

- “dissociative ligand exchange…occurs exceedingly slowly” (rate constant, line 92)

- “associative ligand exchange… can occur very fast” (rate constant, line 94)

4) line 178-183: it would be interesting to mention that similar values in the 100 pM range have been reported in mammalian cells (Qin et al., 2013).

5) section “a brief overview of zinc acquisition and homeostasis in Arabidopsis”: it would be worthwhile stating that all the data published on other plants indicate that the functions of the main players of the homeostatic network revealed in Arabidopsis are conserved in distant species, such as rice.

6) line 243: “This has been attributed” (remove “is”)

---

## [Reviewer Report]

The Manuscript “Changing paradigms of the micronutrient zinc, a versatile and effective protein cofactor and possible signal” focuses on Zn as a micronutrient and its role in plant organisms as a protein cofactor. Although there have been several excellent reviews recently that address Zn homeostasis in plants, including one by the author (Krämer 2024), the main focus has not been on Zn biological chemistry in the context of plant Zn homeostasis. Here, this review systematizes and updates the information available relating Zn chemistry, the different types of interactions between Zn and metalloprotein (Type I-III metalloproteins), and the buffering of Zn in the cellular environment. I think it is an excellent short review, that will be useful and help advancing the field of plant Zn homeostasis and Zn biology.

Comments:

Line 7, title: I think the title might be somehow unclear; I understand and agree that it is important to highlight Zn as a versatile and effective protein cofactor in the field of plant biology, but I don´t think that it represents a change of paradigm (being unclear what the previous paradigm was?). Instead, it is a step forward in providing more systematized information and contextualization on the role of Zn as a micronutrient in plant biology, to which, as I mentioned above, this review is an excellent contribution.

Lines 7 and 190-211: The text focuses on and develops the potential relevance of changes in cellular redox status on Zn homeostasis, suggesting that a condition of oxidative stress could lead to transient increase of intracellular buffered available Zn2+ concentration with impact on signalling and regulation. This is a very interesting and relevant point, which is very well addressed. However, I think that the connection between this point and a role for Zn2+ as a possible signal is not clearly stated. This would be important also in connection with the title, which refers to this possible role for Zn.

Line 208-211: It is referred that a role for Zn2+ (transient variations in buffered available Zn2+) as a signal in plants has hardly received attention, contrary to the animal field. In addition to the references mentioned in these lines, the review by Assuncao 2022 briefly discussed this possibility.

Minor comments:

Pag 61: “There are well-known examples Zn-deficient..” - missing “of”.

Pag 243: “This has been is attributed..” remove “has been/is”.

Pag 265: References are in italic, not consistent.

---

## [Reviewer Report]

In this review article, Ute Kraemer presents an overview of the importance of Zn in Biology due to its particular chemistry, and how this chemistry determines Zn binding and unbinding to ligands. She then describes the Zn functions in plants, Zn homeostasis and buffering mechanisms in plants.

This is a nice piece of work, outlining current challenges and next frontiers in the study of Zn biology in plants. The author is an internationally recognized expert of these questions, and the manuscript is extremely knowledgeable. In this current form, I am not sure if this very expert presentation is fully accessible to a non-expert readership (especially the 1st half of the manuscript, up to ~ line 213). To solve this, I would suggest to take more space to define a number of concepts, maybe in a separate box not to overload the text. Such concepts to be defined could be :

- Irving-Williams series

- dissociative ligand exchange vs associative ligand exchange

- type I, II, III metalloproteins

- lewis acid

- metal buffer and buffered metal concentration. This last one is important, as it is a key concept discussed in the manuscript, and it is not formerly described.

I am wondering if the sections ‘Zn-dependent protein functions in plants’ and ‘Protein metalation, binding and unbinding of cellular Zn2+ ions’ should be inverted. This would connect the Zn chemistry property description to its impact on ligand binding, then allow a connection between Zn function, homeostasis and buffering in plants.

Fig. 1 C and D are very complex and are hard to connect to the text. I would suggest to expand their description in the legend, and to refer to those panels more often in the text where applicable (it is cited only once in the text). It may also require to include some further description of the concepts outlined in the figure in the main text.

The abstract reads like a succession of statements and, at first read, it is not always obvious how these are connected. It could be simplified and restructured to provide a clearer message about the content of the review.

Minor points :

- The formatting of the in-text citations is not consistent (some citations include initials to author first names, whereas most don’t.)

- I would suggest to use the same unit for Zn content in plant and soil (ug/g vs mg/kg)

- The reference Kraemer et al 2024 is heavily cited. Is it always justified vs citing original publications?

---

## [Editor Report]

Dear Dr. Krämer, (liebe Ute),

your manuscript “Changing paradigms of the micronutrient zinc, a versatile and effective protein cofactor and possible signal” has been seen by two independent reviewers. Both are very positive and have only a few minor suggestions that I would like you to take into account in a minor revision.

Thank you for your valuable contribution to the Research Topic “Quantitative approaches to cellular aspects of plant ion homeostasis”.

Best wishes

Ingo

---

## [Editor Report]

Dear Ute,

thank you for the careful revision of the manuscript and thanks again for your valuable contribution to the Research Topic “Quantitative approaches to cellular aspects of plant ion homeostasis”. It is highly appreciated.

Best regards, Ingo